

# Description of the Baseline Surface Radiation Network (BSRN) station at the Izaña Observatory (2009-2017): measurements and quality control/assurance procedures

Rosa Delia García[1,2,3], Emilio Cuevas[2], Ramón Ramos[2], Victoria Eugenia Cachorro[3], Alberto Redondas[2], and José A. Moreno-Ruiz[4]

[1]Air Liquide España, Delegación Canarias, Candelaria, 38509, Spain
[2]Izaña Atmospheric Research Center (IARC), State Meteorological Agency (AEMET), Spain
[3]Atmospheric Optics Group, Valladolid University, Valladolid, Spain
[4]Department of Computer Science, University of Almería, Spain

**Correspondence:** Emilio Cuevas
(ecuevasa@aemet.es)

**Abstract.** The Baseline Surface Radiation Network (BSRN) was implemented by the World Climate Research Programme (WRCP) starting observations with 9 stations in 1992, under the auspices of the World Meteorological Organization (WMO). Currently, 59 BSRN stations submit their data to the WRCP. One of these stations is the Izaña station (Station: IZA, #61) that enrolled in this network in 2009. This is a high-mountain station located in Tenerife (Canary Islands, Spain; at 28.3° N, 16.5° W, 2373 m a.s.l.) and is a representative site of the subtropical North Atlantic free troposphere. It contributes with basic-BSRN radiation measurements, such as, global shortwave radiation (SWD), direct radiation (DIR), diffuse radiation (DIF) and longwave downward radiation (LWD) and extended-BSRN measurements, including ultraviolet ranges (UV-A and UV-B), shortwave upward radiation (SWU) and longwave upward radiation (LWU) and other ancillary measurements, such as vertical profiles of temperature, humidity and wind obtained from radiosonde (WMO, station #60018) and total column ozone from Brewer spectrophotometer. The IZA measurements present high quality standards since more than 98 % of the data are within the limits recommended by the BSRN. There is an excellent agreement in the comparison between SWD, DIR and DIF (instantaneous and daily) measurements with simulations obtained with the LibRadtran radiative transfer model. The root mean square error (RMSE) for SWD is 2.28 % for instantaneous values and 1.58 % for daily values, while the RMSE for DIR is 2.00 % for instantaneous values and 2.07 % for daily values. IZA is a unique station that provides very accurate solar radiation data in very contrasting scenarios: most of the time under pristine sky conditions, and periodically under the effects of the Saharan Air Layer characterized by a high content of mineral dust. A detailed description of the BSRN program at IZA, including quality control and quality assurance activities, is given in this work.

## 1 Introduction

The World Meteorological Organization (WMO) through its Global Change Observing System (GCOS) defined several Essential Climate Variables (ECVs) as physical, chemical or biological variable or a group of linked variables that critically

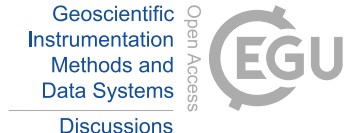


contributes to the characterization of Earth's climate. The ECVs have been selected with the aim to obtain enough evidence that effectively led us to predict the climate evolution and its possible associated risks.

Among others, the surface radiation budget and more specifically the surface Earth Radiation Budget (ERB) longwave and surface ERB shortwave have been identified as ECV, due to its key role in the general circulation of the atmosphere and ocean,

the thermal structure of the atmosphere, and being a main factor in the Earth's climate system (König-Langlo et al., 2013). The surface ERB comprises the fluxes absorbed by the Earth surface and the upward and downward thermal radiative fluxes emitted by the surface and atmosphere, respectively (Myhre et al., 2013).

The first surface solar radiation measurements started in the 1920s in some sites in Europe. The study of these historic datasets reveals an increase in the surface solar radiation until the 1950s, known as *Early Brightening* (Ohmura, 2009; Wild,

2009), but only observed in Europe due to the scarcity of available data.The study of surface solar radiation long-term records show decadal changes with a decline of surface solar radiation from the first available records, around 1950, until the middle of 1980s (Stanhill and Cohen, 2001; Liepert, 2002) period known as *Global Dimming*, and an increment in the surface solar radiation since the middle of 1980s, period known as *Global Brightening* (Wild et al., 2005).

All these studies remark on the variable quality of the data due to the technical advances in the instruments since the 1970s,

thus the confidence of the long-term trends observed should be taken into account when analysing the results. With the aim to obtain data with the best possible quality, in the 1990s, efforts were made to establish measurement networks around the Earth with high quality requirements to avoid introducing undesirable uncertainties in the long-term series. In this context, the Baseline Surface Radiation Network (BSRN) was proposed in 1980 by the WMO and created in 1992 to provide accurate irradiances at selected sites around the Earth, with a high temporal resolution. The BSRN is a project of the World Climate Research Pro-

gramme (WCRP). In 2004 it was designated as the baseline network for GCOS. The available data covers a period from 1992 to the present thanks to the contribution of 59 stations covering various climate zones (http://bsrn.awi.de/nc/stations/maps/). The BSRN data have been widely used due to its quality and reliability in the validation of satellite observations, as input to climate models and to monitor the solar radiation reaching the Earth Surface (Ohmura, 2009).

The BSRN imposes very strict measurement requirements in order to assure the required quality of data (Long and Dutton,

2002; Long and Shi, 2006). A BSRN site must be representative of the surrounding area, avoid pollution sources, unnatural reflectance, microclimate conditions, and human activities that can affect its representativity of the surroundings (McArthur, 2005). Consequently, BSRN sites cannot be located near to major roadways, airports, vehicle parking areas, and buildings.

In 2009 the Izaña Atmospheric Observatory (IZA, BSRN station no: #61) started its process to become a BSRN station through a specific agreement between the State Meteorological Agency of Spain (AEMET) and the University of Valladolid.

IZA was proposed and accepted to be part of the BSRN at the 11[th] BSRN Workshop and Scientific Review meeting was held in Queenstown, New Zealand in August 2012 (WRCP, 2012), being since then a member of the network without interruptions.

Between 2013 and 2014, the UV-A and UV-B radiation measurements performed at IZA were used for satellite-based data validation. The validation resulted in a good agreement with satellite-based data, and it's the starting point for further developments of Flyby's elaboration processes (WRCP, 2014).





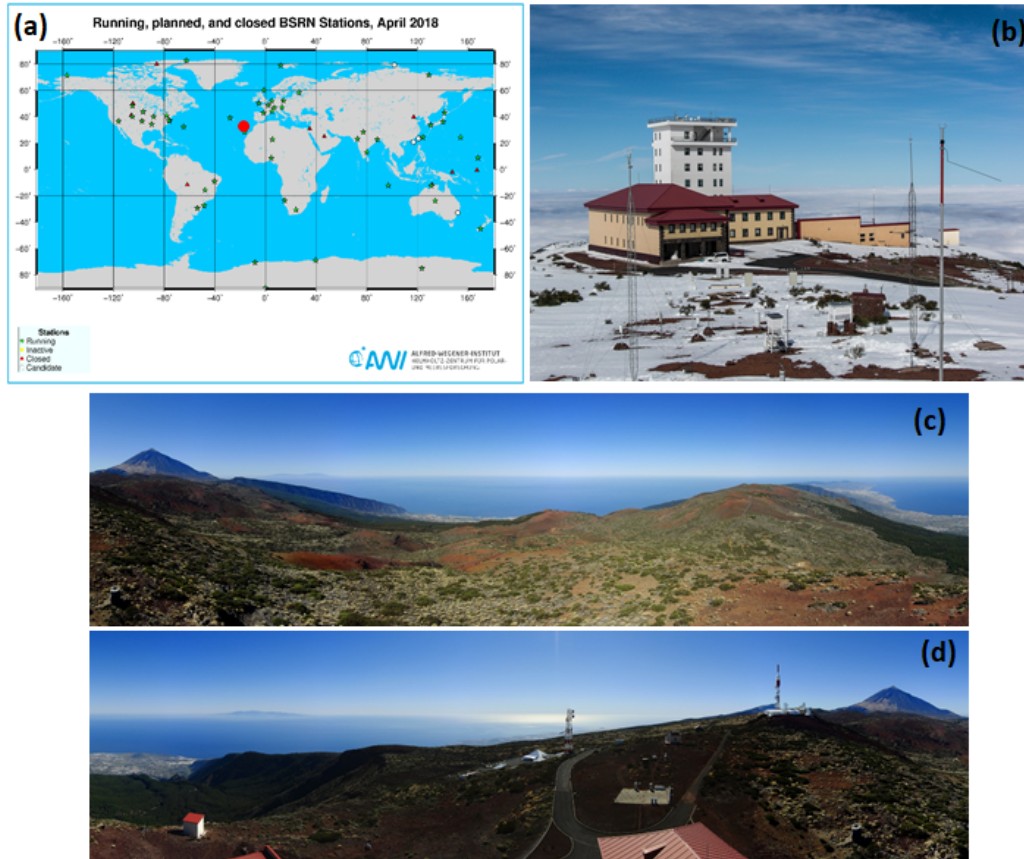

**Figure 1.** (a) Location of the Izaña station (IZA) on a global map of all BSRN stations (http://bsrn.awi.de). (b) Izaña Atmospheric Observatory. Views of Izaña radiation station: (c) Northern and Eastern Views (Azimuth 360, Inclination 0° degrees; Azimuth 90°, Inclination 0°, respectively) (d) Southern and Western Views (Azimuth 180°, Inclination 0° and Azimuth 270°, Inclination 0°, respectively).

The main goal of this work is to present the status of the Izaña BSRN (IZA) between 2009 and 2017. Sect. 2 describes the IZA site. The main characteristics of the instruments and measurements that are part of IZA BSRN as well as instrument calibrations are presented in Sect. 3. Sect. 4 illustrates data processing and quality control procedures applied to the measurements, and the shipment station-to-BSRN archive. Finally, summary and conclusions are given in Sect. 6.

## 2    Site Description

IZA station (http://izana.aemet.es) is managed by the Izaña Atmospheric Research Center (IARC) and is part of AEMET. It is located on the island of Tenerife (Canary Islands, Spain; at 28.3° N, 16.5° W, 2373 m a.s.l.) (Figure 1).

IZA is a high-mountain station above a quasi-permanent strong temperature inversion layer that prevents the arrival of local pollution from lower levels of the island. This meteorological feature favours measurements under free troposphere conditions

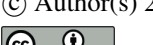



(Cuevas et al., 2013). As a result, the climate in the area of the station is extremely dry for the majority of the year, this together with clean air from middle/upper troposphere gives to the area a high scientific interest. Pristine conditions are alternated with periodical intrusions of the dust-laden Saharan Air layer (Cuevas et al., 2015, 2017; Rodríguez et al., 2015), mainly in summertime. IZA registers the highest average annual insolation duration of Spain with about 3473 h of sunshine per year and

5 an average of 179.5 days/year of clear-days in the climate period 1981-2010 (for more information see http://www.aemet.es).

IZA enrolled in the WMO Global Atmosphere Watch (GAW) programme in 1989. In addition, IZA has contributed to several international networks such as NDACC (Network for the Detection of Atmospheric Composite Change, http://www.ndsc.ncep.noaa.gov) since 1999, and GAW-PFR (Precision Filter Radiometer Network, http://www.pmodwrc.ch/worcc) since 2001. In 2003, the WMO/GAW Regional Brewer Calibration Centre for Europe (RBCC-E, www.rbcc-e.org) was established at IZA. IZA has

10 been part of the Aerosol Robotic Network (AERONET, http://aeronet.gsfc.nasa.gov) since 2004, as one of the two absolute AERONET calibration sites. IZA is also a BSRN station since 2009 (Cuevas-Agulló, 2017). Moreover, in 2014, IZA was appointed by WMO as a CIMO (Commission for Instruments and Methods of Observation) Testbed for aerosols and water vapor remote sensing instruments (WMO, 2014). A detailed description of the IZA site and its observation programs can be found in Cuevas et al. (2017).

## 15  3   Measurements and Instruments

### 3.1   Basic-BSRN Measurements

The basic-BSRN measurements of the BSRN Program at IZA are global shortwave radiation (SWD), direct radiation (DIR), diffuse radiation (DIF) and longwave downward radiation (LWD) (Table 1).

At present, SWD and DIF are measured with unshaded and shaded EKO MS-802F pyranometers (Figure 2a and 2b) (ISO-
9060 classification: secondary standard), respectively. This pyranometer is a high precision instrument with a spectral range from 285 to 3000 nm with a response time less than 5 s. (95 %, confidence level), and an expected uncertainty < $\pm$ 1% for daily totals. The BSRN accuracy target for DIF and SWD is 2 % (5 W m$^{-2}$) and 2 % (3 W m$^{-2}$), respectively (McArthur, 2005).

DIR is measured with an EKO MS-56 pyrheliometer (Figure 2d) (ISO-9060 classification: first-class). This instrument has a full operating view angle of 5° and slope angle of 1°. The spectral range covers from 200 to 4000 nm (50 % points) with a
response time > 1 s. (95 %). The expected uncertainty is < $\pm$ 1 % for daily totals. The BSRN accuracy target for DIR is 0.5 % (1.5 W m$^{-2}$) (McArthur, 2005). LWD is measured with a shaded Kipp & Zonen CGR4 pyranometer (Figure 2c) (ISO-9060 classification: secondary standard). The spectral range is 4.5-42 $\mu$m (50 % points) with a response time less than 6 s. (63 % response). The expected uncertainty is < 3 % for daily totals (95 %). The BSRN accuracy target for LWD is 2 % (3 W m$^{-2}$) (McArthur, 2005).

These instruments are installed on a sun tracker, except the EKO MS-802F pyranometers for SWD and DIF measurements that are installed on a horizontal table (Figure 2a). The sun tracker is an Owel INTRA 3 (Figure 2e). This is an intelligent tracker which combines the advantages of automatic-tracking operation (automatic alignment with the system of astronomical coordinates), and actively-controlled tracking (a 4-quadrant sun sensor). It is constructed for use under extreme weather conditions;





**Table 1.** Basic-BSRN radiation instruments installed between 2009 and 2017 at IZA BSRN. (SWD, DIR, DIF and LWD). The instruments currently in operation are marked in bold.

| Parameter | Manufacturer | Type | Serial Number | WRCM | Starting Date | Finish Date | Spectral Range |
|---|---|---|---|---|---|---|---|
| **SWD** | Kipp & Zonen | CM-21 | 080034 | 61001 | 01/03/2009 | 10/11/2016 | 335-2600 nm |
| | **EKO** | **MS-802F** | **F15509FR** | **61010** | **11/11/2016** | **—-** | **285-3000 nm** |
| **DIR** | Kipp & Zonen | CH-1 | 080050 | 61003 | 01/03/2009 | 10/11/2016 | **200-4000 nm** |
| | **EKO** | **MS-56** | **F15048** | **61012** | **11/11/2016** | **—-** | |
| **DIF** | Kipp & Zonen | CM-21 | 080032 | 61002 | 01/03/2009 | 10/11/2016 | **335-2600 nm** |
| | **EKO** | **MS-802F** | **F15508FR** | **61011** | **11/11/2016** | **—-** | |
| **LWD** | **Kipp & Zonen** | **CGR-4** | 080022 | 61004 | 01/03/2009 | 01/05/2009 | **4.5-42 $\mu$m** |
| | | | 050783 | 61008 | 01/05/2009 | 13/05/2014 | |
| | | | 080022 | 61004 | 14/05/2014 | 22/07/2014 | |
| | | | 050783 | 61008 | 23/07/2014 | 30/03/2017 | |
| | | | 080022 | 61004 | 30/03/2017 | 07/06/2017 | |
| | | | **050783** | **61008** | **08/06/2017** | **—-** | |

its operational temperature range is between -20 and 50° C. It can sustain about 50 kg of carefully balanced load. The tracker motors have a special grease for use in low temperatures. It moves back to the start (morning) position at the corresponding midnight. The drive unit has a zenith rotation > 90°. The unit has an angular resolution $\leq$ 0.1°, an angular repeatability of $\leq$ $\sim$ 0.05 ° and an angular velocity $\geq$ 1.5 °/s on the outgoing shafts. The maximum speed is 2.42 °/s (Georgiev et al., 2004).

5    In addition, the measurements of pressure (P), relative humidity (RH) and temperature (T) are included in this measurement group. The pressure is measured a with Setra 470 Pressure Transducer and RH and T are measured with Campbell Scientific CS215-L sensors.

## 3.2    Extended-BSRN Measurements

The extended-BSRN measurements included in the IZA BSRN program are shortwave upward radiation (SWU), ultraviolet measurements (UV-A and UV-B), and longwave upward radiation (LWU) (Table 2).

10    A Yankee YES pyranometer (Figure 3a) measures global radiation in the UV-B spectral range from 280 to 315 nm with a response time around 100 ms. The UV-A (315-400 nm) is measured with a Kipp & Zonen UV-A-S-T pyranometer (Figure 3b) with a response time less than 1.5 s. (95 %). The expected uncertainty is < 5 % for daily totals (95 %). SWU and LWU are measured with a MS-60 EKO radiometer (Figure 3c) (ISO-9060 classification: secondary standard). This system is formed by two pyranometers and two pyrgeometers. The spectral range of the pyranometers is 280-3000 nm with a response time $\sim$ 17 s, while the spectral range of the pyrgeometers is 3-50 $\mu$m.

The radiation measurements are acquired with a Campbell CR5000. This datalogger is a rugged, high-performance data-acquisition system with built-in keyboard, graphics display, and PCMCIA card slot. It combines 16-bit resolution with a





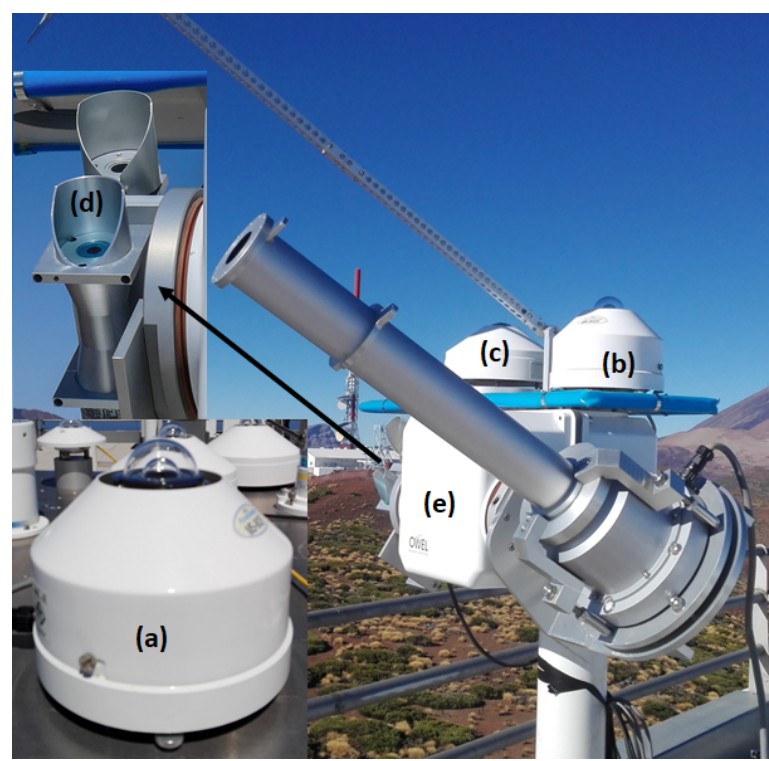

**Figure 2.** Basic-BSRN radiation instruments currently in operation at IZA BSRN. (a) SWD: EKO MS-802F pyranometer installed on a table horizontal; (b) DIF: EKO MS-802F pyranometer; (c) LWD: Kipp & Zonen CGR4 pyrgeometer; (d) DIR: EKO MS-56 pyrheliometer and; (e) sun tracker: Owel INTRA 3.

**Table 2.** Extended-BSRN radiation instruments installed between 2009 and 2017 at IZA BSRN. (UVB, UVA, SWD and LWU).Same as Table 1

| Parameter | Manufacturer | Type | Serial Number | WRCM | Starting Date | Finish Date | Spectral Range |
|---|---|---|---|---|---|---|---|
| **UV-B** | **Yankee YES** | **UVB-1** | 970839 | 61007 | 01/03/2009 | 22/02/2010 | **280-315 nm** |
| | | | 071221 | 61009 | 22/02/2010 | 22/07/2015 | |
| | | | **970839** | **61007** | **23/07/2015** | —- | |
| **UV-A** | **Kipp & Zonen** | **UV-A-S-T** | **08005** | **61006** | **01/03/2009** | **—-** | **315-400 nm** |
| **SWD and LWU** | Kipp & Zonen | CRN1 | 030693 | 61005 | 01/03/2009 | 27/11/2016 | PYR: 305-2800 nm PYRG: 5-50 $\mu$m |
| | **EKO** | **MR-60** | **S15115.07** | **61013** | **01/01/2017** | **—** | **PYR: 285-3000 nm PYRG: 3-50 $\mu$m** |





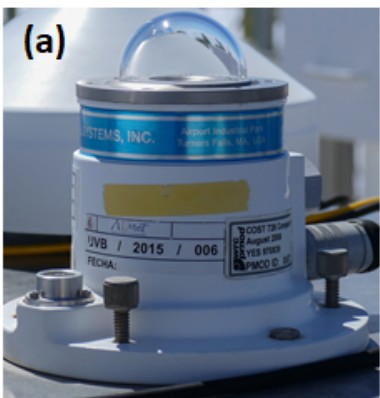
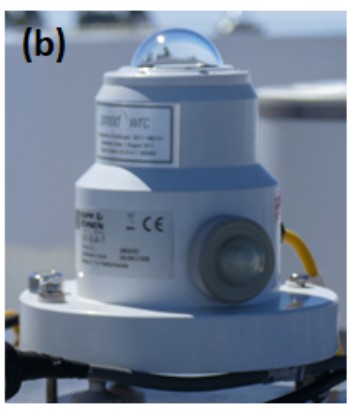
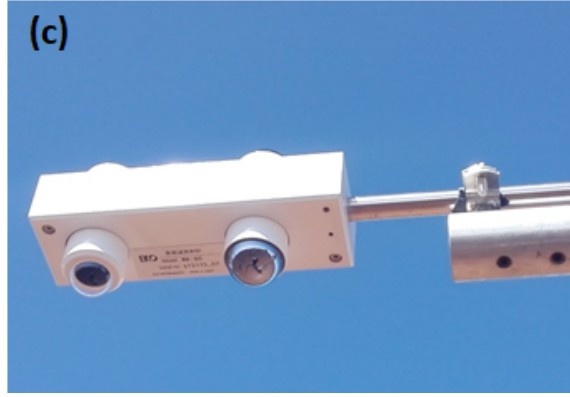

**Figure 3.** Extended-BSRN radiation instruments currently in operation at IZA BSRN. (a) UV-B: Yankee YES pyranometer; (b) UV-A: Kipp & Zonen UV-A-S-T pyranometer and; (c) SWU and LWU: EKO M-60 net radiation sensor.

maximum of 5000 measurements per second. In particular, the measurements are taken with a time step of 5 s. The minimum, average, maximum and standard deviation are stored every minute.

### 3.3 Ancillary Measurements

Ancillary measurements are performed at IZA BSRN station such as radiosonde data (Figure 4a) and total ozone column
(TOC).

Vertical profiles of pressure, temperature, relative humidity and wind direction and speed are measured using Vaisala RS92 radiosondes (Carrillo et al., 2016; Cuevas et al., 2017) that are launched twice a day, at 00 and 12 UTC at the Güimar station (WMO GRUAN station #60018, 105 m a.s.l.), managed by the Meteorological Centre of Santa Cruz de Tenerife (AEMET). This station is located near the coastline at a distance in a straight line from IZA of ∼15 km. The TOC measurements are
10 performed with Brewer spectrophotometer (Figure 4b) (precision better than 1 %) (Redondas and Cede, 2006). An automatic Cloud Observation System (SONA camera) (Figure 4c) (González et al., 2013) developed by Sieltec Canarias S.L. takes all-sky images every 5 minutes, day and night. This camera consists of a 640x480 pixels resolution, 8 bit color response CCD sensor with Bayer filter, with spectral range from 400 to 700 nm. A rotating shadow band is used for protecting the sensor from direct sunlight.

### 3.4 Instrument checks and maintenance

All the instruments of the BSRN are checked on a daily basis by meteorological observers of the Izaña observatory. Routine checks consist of cleaning the domes, cable connections inspection and instrument levelling, as well as, checking the proper functioning of the solar tracker and shading system of the instruments for DIF and LWD measurements. Recently, a tool to test the Owel INTRA solar tracker real time check-up has been implemented. This test consists in controlling the 4 quadrants of





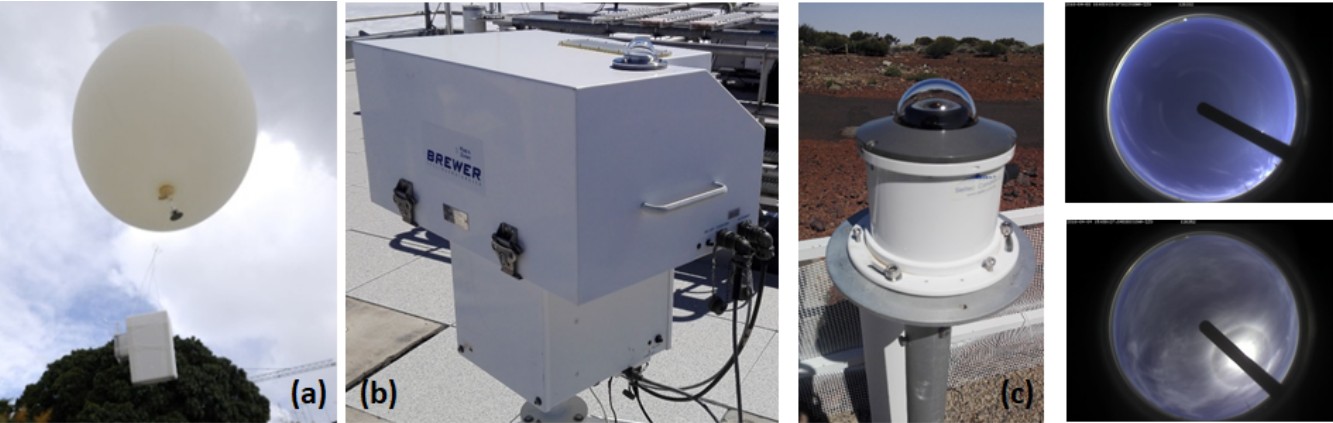

**Figure 4.** Other instruments currently providing data to IZA BSRN (a) radiosonde profiles, (b) Brewer spectrophotometer and (c) SONA camera (Automatic Cloud Observation System) installed at IZA and examples of images taken by the SONA camera.

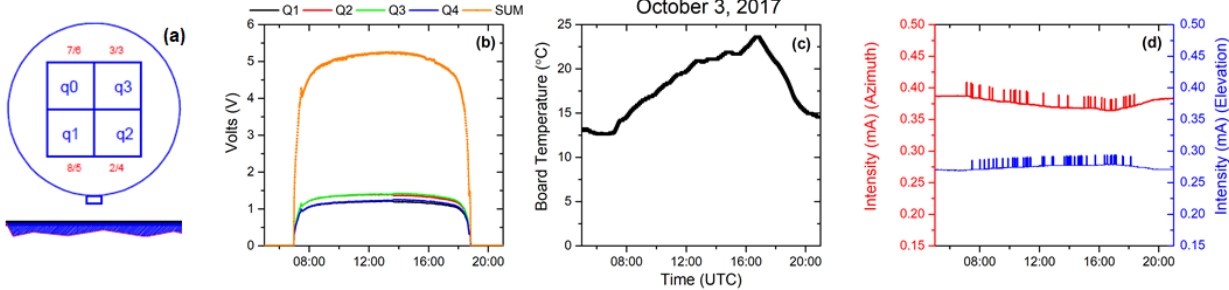

**Figure 5.** Daily control test of the Owel INTRA solar tracker at IZA. (a) Front view of the quadrants of the sun detector. The numbers indicated are pins of the sensor/connector resp; (b) sun-sensor signals in the four quadrants and the total signal from the quadrants; (c) Board temperature (°C) and; (d) current from base-shunt of motor0-driver (Azimuth axis, mA) (red color) and current from base-shunt of motor1-driver (Elevation axis, mA) (black color) (INTRA, 2010).

the solar tracker (see Sect. 3.1), checking its levelling, the board temperature and the intensity of the base-shunt of motor in azimuth and elevation axis (Figure 5).

### 3.5 Instrument Calibrations

All the radiation instruments (Table 1 and 2) have been periodically calibrated following the recommendations of the BSRN
5 (Table 3) and are regularly compared with reference instruments with recent calibration from the World Radiation Center (WRC) at Davos.

An Absolute Cavity Pyrheliometer PMO6 designed at PMOD (Physikalisch-Meteorologisches Observatorium Davos) (Figure 6) that is regularly calibrated at the WRC, is used as reference instrument and is directly traceable to the World Radiometric



**Table 3.** Summary of calibrations of the different radiation instruments performed at IZA between 2009 and 2017.(*) (W m$^{-2}$) V-1.

| Parameter | Manufacturer | Serial Number | Calibration Date | Calibration Site | Calibration Factor ($\mu$V/W m$^{-2}$) |
|---|---|---|---|---|---|
| **SWD** | Kipp & Zonen CM-21 | 080034 | 27/03/2008 | Factory Calibration Kipp & Zonen | 8.86 |
| | | | 18/07/2014 | IZA/AEMET | 8.67 |
| | EKO MS-802F | F15509FR | 26/04/2016 | Factory Calibration EKO | 7.04 |
| **DIR** | Kipp & Zonen CH-1 | 080050 | 19/06/2008 | Factory Calibration Kipp & Zonen | 9.98 |
| | | | 18/07/2014 | IZA/AEMET | 9.91 |
| | EKO MS-56 | F15048 | 26/01/2016 | Factory Calibration EKO | 8.99 |
| **DIF** | Kipp & Zonen CM-21 | 080032 | 27/03/2008 | Factory Calibration Kipp & Zonen | 8.63 |
| | | | 18/07/2014 | IZA/AEMET | 8.68 |
| | EKO MS-802F | F15509FR | 26/04/2016 | Factory Calibration EKO | 7.05 |
| **LWD** | Kipp & Zonen CGR-4 | 080022 | 28/02/2008 | Factory Calibration Kipp & Zonen | 10.37 |
| | | | 12/01/2007 | Factory Calibration Kipp & Zonen | 9.77 |
| | | 050783 | 30/06/2014 | PMOD/WRC | 9.39 |
| | | | 27/03/2017 | PMOD/WRC | 9.41 |
| **UV-B** | Yankee YES UVB-1 | 071221 | 28/02/2008 | Factory Calibration Yankee YES | 199 (*) |
| | | 970839 | 23/08/2006 | PMOD/WRC | 0.1178 (*) |
| | | | 12/07/2015 | AEMET | 0.1191 (*) |
| **UV-A** | UV-A-S-T | 080005 | 27/10/2006 | Factory Calibration Kipp & Zonen | 32.469 (*) |
| | | | 08/08/2017 | PMOD/WRC | 30.98 (*) |
| | Kipp & Zonen | 030693 | 08/01/2003 | Factory Calibration Kipp & Zonen | 10.09 |
| **SWU and LWU** | EKO MR-60 | S15115.07 | 12/05/2016 | Factory Calibration EKO | SWU:6.95; SWD:6.8 LWU:3.08; LWD:3.22 |




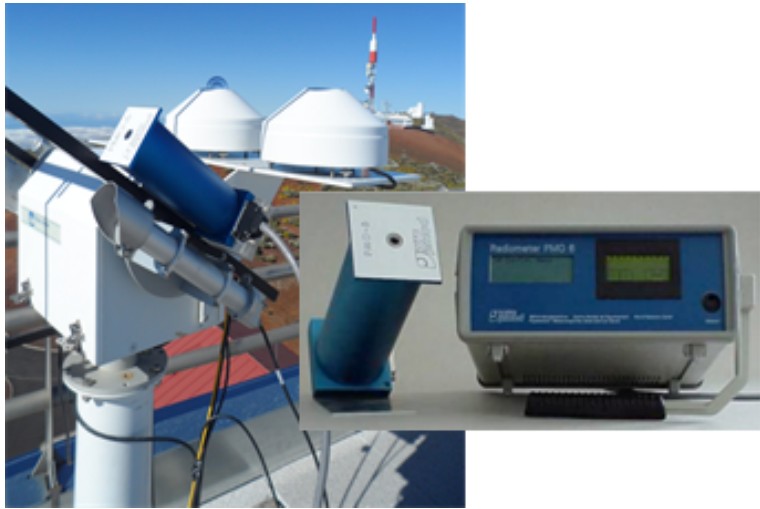

**Figure 6.** Absolute cavity radiometer (PMO6) mounted on the IZA BSRN sun-tracker during a calibration campaign.

Reference. Periodical calibrations with PMO6 allow us to assure the reliability of the measurements and correct time degradation on the calibration constants. A large calibration campaign of BSRN pyranometers and pyrheliometers was performed during 2014 using the fore mentioned PMO6. The ISO 9059:1990 (E) and ISO 9846:1993(E) recommendations were met. The calibration of a field pyranometer/pyrheliometer by means of a reference pyrheliometer is accomplished by exposing the two

instruments to the same solar radiation and comparing their corresponding measurements. This allows us to compare target instruments to high accuracy radiation sensors. A second calibration campaign was held in July-August of 2018.

## 4   BSRN IZA management

During 2009 a BSRN database was developed in order to manage the large volume of BSRN data. This tool not only allows the management of a large volume of information automatically generated, but it is also used for checking of real-time

measurements becoming a comprehensive quality control system with corresponding alarms.

The BSRN data management flowchart is shown in Figure 7. It includes daily and monthly semi-automatic processes to collect and check the measurements, and generate the station-to-archive file sent every month to the BSRN database. This daily process can be executed automatically, or on demand, producing several warning alerts if human checking is needed.

Data are stored in a CR5000 datalogger (see Sect. 3.2). This datalogger generates a *Raw data* file that is stored in a database

for further analysis, if necessary, which is also available on an internal web for real-time access. The *Raw data* file is checked in order to assess the format integrity and detect gaps.

### 4.1   Measurements Radiation Corrections

Several corrections are applied to raw data to obtain the final radiation data. These corrections are:



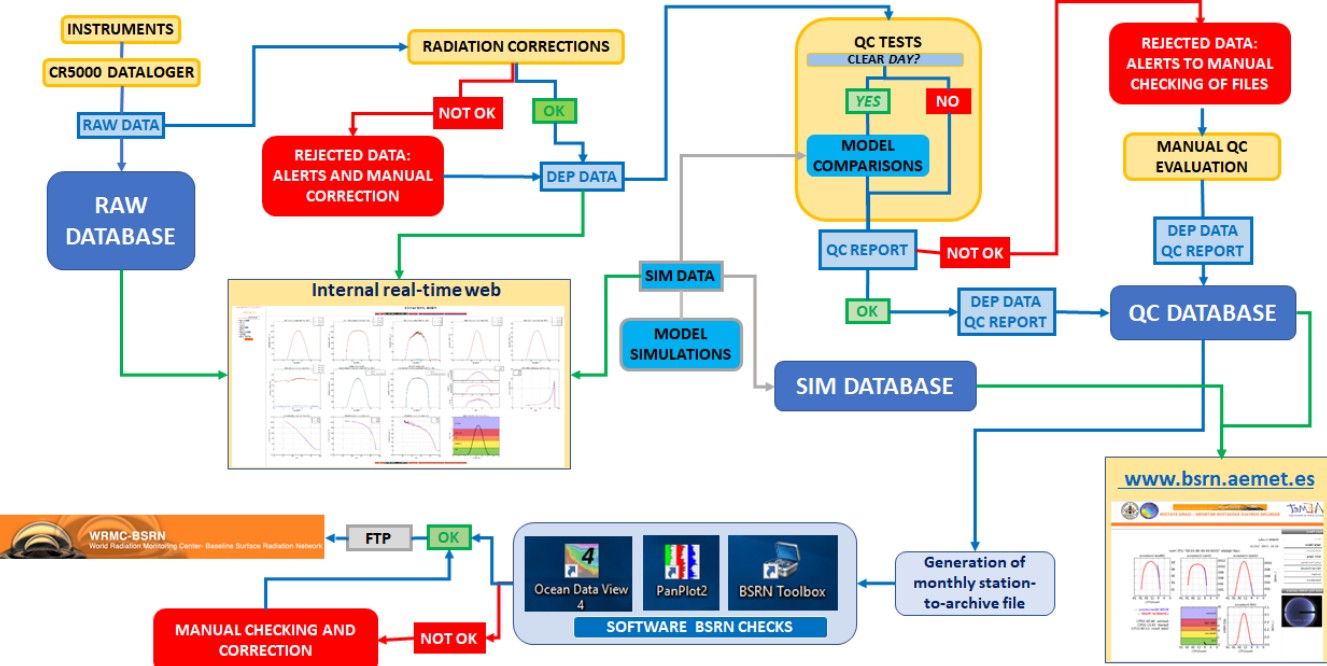

**Figure 7.** Flowchart of data from measurements to the BSRN database

– **Zero offset:** defined as the signal caused by changes in the instrument temperature. The zero offset is measured for each instrument as part of the observation sequence when possible. For instruments that are not capable of obtaining a zero offset with each observation, it is measured at night and subtracted from daytime values (McArthur, 2005). The average values of zero offset compared to the radiation values performed during the day are rather small, representing 0.3 % and

0.02 % of the SWD and DIR signals for 1000 W m$^{-2}$, respectively (García Cabrera, 2011).

– **Cleaning operations:** as remarked in Sect. 3.4, daily cleaning of domes is performed. Some artificial shadows are caused when the observers perform these operations. Data corresponding to cleaning activities are identified and removed from the database (Figure 8).

– **Exceptional situations:** shadows or gaps in raw data are also observed due to exceptional situations, such as severe

weather, repairing of instruments and maintenance operations, etc. Data stored during these non-operational periods are also removed from the database.

## 4.2   Quality Control (QC)

Once the corrections are made a Dep data file is obtained, that will be used to perform the quality control (QC) tests. The IZA QC procedure has two main parts, the recommended BSRN controls, and the comparison with simulations with radiative

transfer models (RTMs).

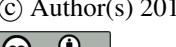


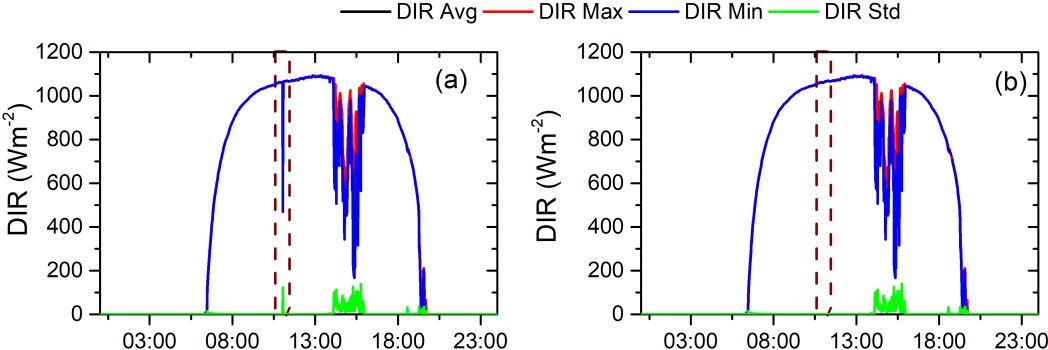

**Figure 8.** Example of data removing process. The rejected data were caused by shadows during cleaning operations. The dashed line represents the pyrheliometer cleaning time; a) before removing rejected data and; b) after removing rejected data.

**Table 4.** The lower and upper limits for the "physically possible" (PPL) and "extremely rare" (ERL) limits used in flagging the radiation measurements. $\mu_o$ is the cosine of the solar zenith angle and $S_a$ is the solar constant adjusted for the Earth-Sun distance.

| | LOWER BOUND (W m$^{-2}$) | | UPPER BOUND (W m$^{-2}$) | |
|---|---|---|---|---|
| Parameter | PPL | ERL | PPL | ERL |
| SWD | -4 | -2 | $S_a*1.5*\mu_o^{1.2}+100$ | $S_a*1.2*\mu_o^{1.2}+50$ |
| DIF | -4 | -2 | $S_a*0.95*\mu_o^{1.2}+50$ | $S_a*0.75*\mu_o^{1.2}+30$ |
| DIR | -4 | -2 | $S_a$ | $S_a*0.95*\mu_o^{0.2}+10$ |
| SWU | -4 | -4 | $S_a*1.2*\mu_o^{1.2}+50$ | $S_a*\mu_o^{1.2}+50$ |
| LWD | 40 | 60 | 700 | 500 |
| LWU | 40 | 60 | 900 | 700 |

The first part of the QC consists of applying the QC methods that the WRMC recommends to the BSRN data (Gilgen et al., 1995; Ohmura et al., 1998; Long and Dutton, 2002; Long and Shi, 2006, 2008). These quality control procedures are based on checking whether the measurements are within certain limits: physically possible limits (PPL), extremely rare limits (ERL), and the comparison of various irradiance components.

5    The PPL procedure is introduced for detecting extremely large errors in radiation data, while the ERL procedure is used to identify measurements exceeding the extremely rare limit. Radiation data exceeding these limits normally occur under very rare conditions and over very short time periods. These tests are based on empirical relations of different quantities (Table 4).

The final BSRN QC procedure is the comparison of various radiation components, i.e., the ratio between the DIR, directly measured with a pyrheliometer, and the derived value from the difference between the SWD and DIF (SumSWD), and the

10    ratio between DIF and SWD. These tests capture smaller errors that have not been detected by the PPL and ERL procedures (Table 5).





**Table 5.** Same as Table 4 except for "comparisons" intervals used for flagging the radiation quantities. $\sigma$: Stephan-Boltzmann constant $(5.67\times10^{-8}$ W m$^{-2}$ K$^4$), SZA: solar zenith angle, T: air temperature (K) and SumSWD = DIR*cos(SZA) + DIF.

| Comparison | Conditions | Test |
|---|---|---|
| **SWD/SumSWD** | SZA<75° <br> SumSWD > 50 W m$^{-2}$ | SWD/SumSWD $\sim \pm$ 8 % |
| | 75°<SZA<93° <br> SumSWD > 50 W m$^{-2}$ | SWD/SumSWD $\sim \pm$ 15 % |
| | SumSWD < 50 W m$^{-2}$ | Test not possible |
| **DIF/SWD** | SZA<75° <br> SumSWD > 50 W m$^{-2}$ | DIF/SWD < 1.05 |
| | 75°<SZA<93° <br> SumSWD > 50 W m$^{-2}$ | DIF/SWD < 1.10 |
| | SWD < 50 W m$^{-2}$ | Test not possible |
| **LWD to Air Temperature** | 0.4 $\sigma$ T$^4$ < LWD < $\sigma$ T$^4$ + 25 W m$^{-2}$ | |

At IZA, the measurements quality assessment is performed taking into account the tests described above, by using the software "BSRN ToolBox" (Schmithüsen et al., 2012) developed for the BSRN community and WRCM. This software also includes a data format check for the station to archive files, and for PANGAEA download files (see below). Data quality checks as outlined in the "BSRN Global Network Recommended QC tests, V2.0" (Long and Dutton, 2002) can also be performed with this software. We have found that < 1 % of all radiation measurements at IZA are outside the PPL and ERL limits (see Figure 9a and 9b) between 2009 and 2017 for solar zenith angles (SZA) < 90 °.

The ratio between the different components also confirms the high quality of the SWD, DIR, and DIF measurements. For SWD/SumSWD and SZA < 75 ° > 98 % (Figure 9c) of the data are between 0.92 and 1.08, while for 75 ° < SZA < 93 °, 96 % of the measurements range from 0.85 to 1.15 (Figure 9d). For DIF/SWD the results present a high quality with 99 % within the established limits, for both SZA < 75 ° and 75 ° < SZA < 93 ° (Figure 9e and 9f). The IZA radiation measurements largely meet the BSRN quality controls.

The second part of the QC is the comparison of instantaneous and daily radiation measurements with simulations performed with RTMs during clear periods. An adaption of the Long and Ackerman's method (Long and Ackerman, 2000) for IZA, performed by García et al. (2014), is used for detecting instantaneous clear-sky periods. This method is based on 1-minute SWD and DIF measurements to which four individual tests are applied to normalized SWD, maximum DIF, change in SWD with time and normalized DIF ratio variability.

Following the BSRN recommendations, the instantaneous clear-sky periods detected are simulated and compared with instantaneous and daily radiation measurements. The RTM model used is LibRadtran (http://www.libradtran.org; Mayer and Kylling (2005); Emde et al. (2016)) that has been extensively tested at IZA (García et al., 2014; García et al., 2018). The model




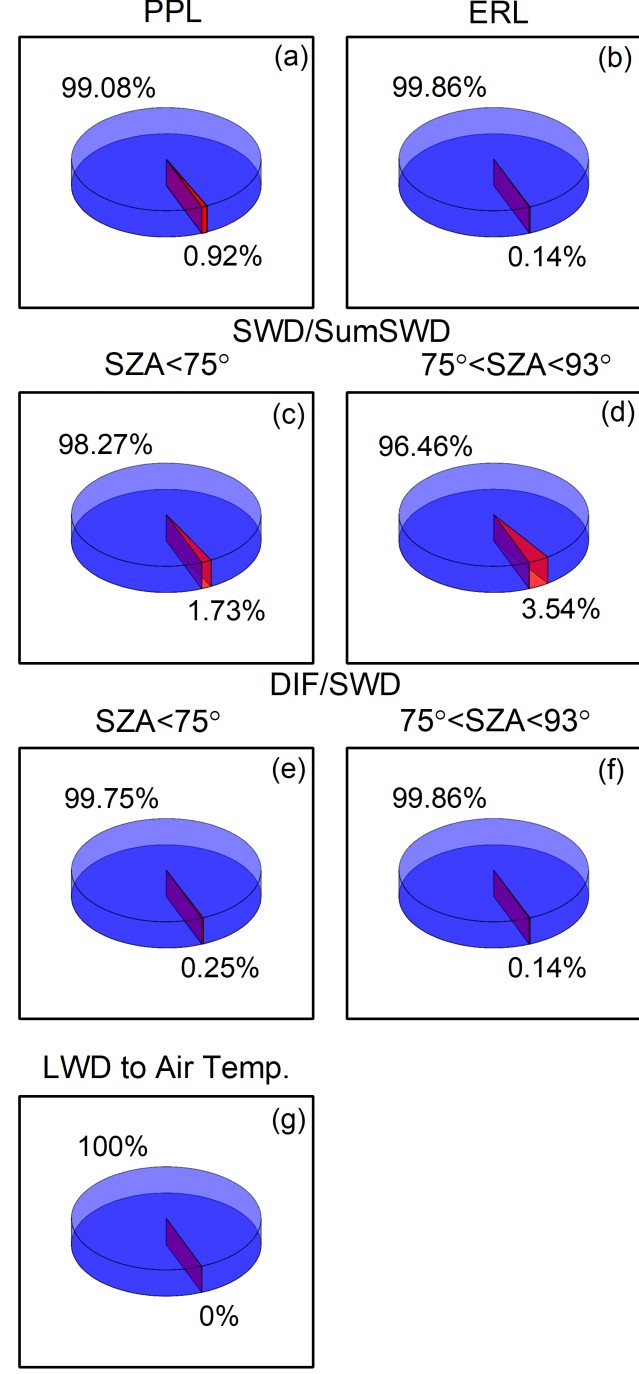

**Figure 9.** Percentage of rejected data (red color) and accepted data (blue color) according to the physically possible limits (PPL) and extremely rare limits (ERL), and comparison of various irradiance components (SWD, SumSWD, DIF, LWD and air temperature) at IZA between 2009 and 2017.




**Table 6.** Statistics for the bias between instantaneous and daily SWD, DIR and DIF simulations and measurements at IZA BSRN in the period 2009-2017. MB, mean bias; STD, standard deviation and RMSE, root mean square error.

| | MB (%) | | STD (%) | | RMSE (%) | |
|---|---|---|---|---|---|---|
| | **Instantaneous** | **Daily** | **Instantaneous** | **Daily** | **Instantaneous** | **Daily** |
| **SWD** | -1.68 | -1.24 | 2.26 | 1.03 | 2.28 | 1.58 |
| **DIR** | -1.57 | -1.82 | 1.92 | 1.17 | 2.00 | 2.07 |
| **DIF** | 0.08 | 0.84 | 7.90 | 8.69 | 9.89 | 9.11 |

input parameters (precipitable water vapor (PWV), aerosol optical depth (AOD), total ozone column and surface albedo) are measured at IZA (García et al., 2014; García et al., 2018). The straightforward comparison between the instantaneous and daily SWD, DIR and DIF simulations and measurements shows an excellent agreement (Figure 10). The variance of daily (instantaneous) measurements overall agrees within 99 % (98 %) of the variance of daily (instantaneous) simulations.

The simulations slightly underestimate the instantaneous/daily measurements of SWD (-1.68 % / -1.24 %) and DIR (-1.57 % / -1.82 %), while the DIF simulations overestimate the instantaneous/daily measurements (0.08 % / 0.84 %). The RMSE is < 2.5 % for SWD and DIR for both instantaneous and daily comparisons, while for DIF it increases to 9.89 % and 9.11 % for instantaneous and daily comparisons, respectively (Table 6). These results are in agreement with those obtained by García et al. (2014).

**4.3  Web-Tool**

With the aim to have, at a glance, an overview of the state of the BSRN station, a web site has been developed for the IZA BSRN station (Figure 11; http://www.bsrn.aemet.es). Plots for several variables such as SWD, DIR, DIF, UV-B, UVI index are automatically available at the home web page, as well as corresponding simulations performed with actual input data at IZA station. These plots are provided in near-real-time (every 10 minutes).

In the webpage there are links to the comparison between measurements and simulations, QC control results, long-term series and derived products, among other additional information. Additional information on the installed instrumentation and the BSRN-related publications is also available (Figure 12). In the following paragraphs we present a short description of the BSRN Izaña, long-term series and derived products.

– **BSRN Izaña:** In this menu, it is possible to select any date and plot the results of applying the QC recommended by
the BSRN (see Sect. 4.2) for SWD, DIR and DIF. It is also possible to plot the comparison of measured and simulated SWD, DIR, DIF and UV-B radiation at IZA using LibRadtran RTM and input parameters measured at IZA. This section of the web is automatically updated every night, once the measured input parameters for the model are available, and the QC tests are applied, according to the flowchart shown in Figure 7.

– **Long-term series:** Daily values of SWD, DIR, DIF, UV-A and UV-B measurements and instantaneous (11 UTC) values
of LWD are correspondingly updated on the web (Figure 13). SWD measurements started in 1977 with a bimetallic





**Figure 10.** Scatterplots and histograms of the instantaneous (W m$^{-2}$) and daily (MJ m$^{-2}$) radiation measurements and simulations for the period 2009-2017: (a),(d),(g) and (j), SWD (b),(e),(h) and (k) DIR and (c),(f),(i) and (l) DIF. The fitting parameters are shown in the legend.





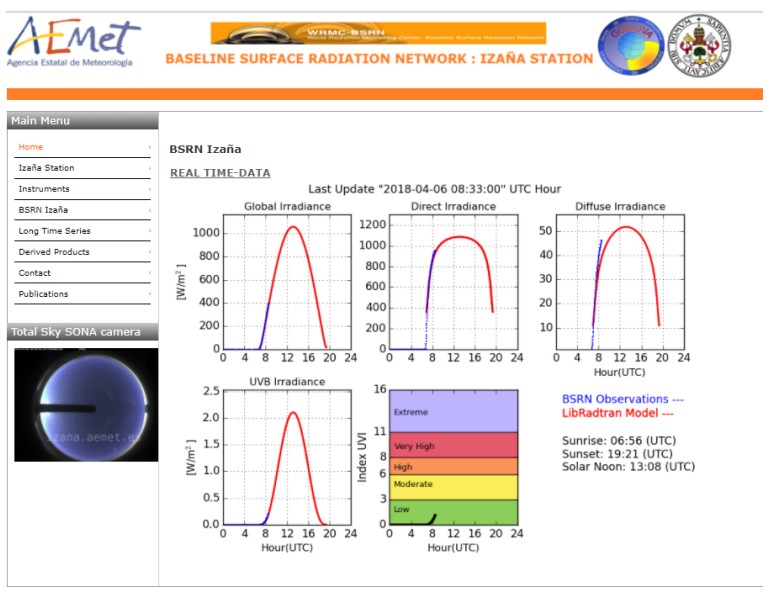

**Figure 11.** Baseline Surface Radiation Network: Izaña Station homepage (http://www.bsrn.aemet.es).

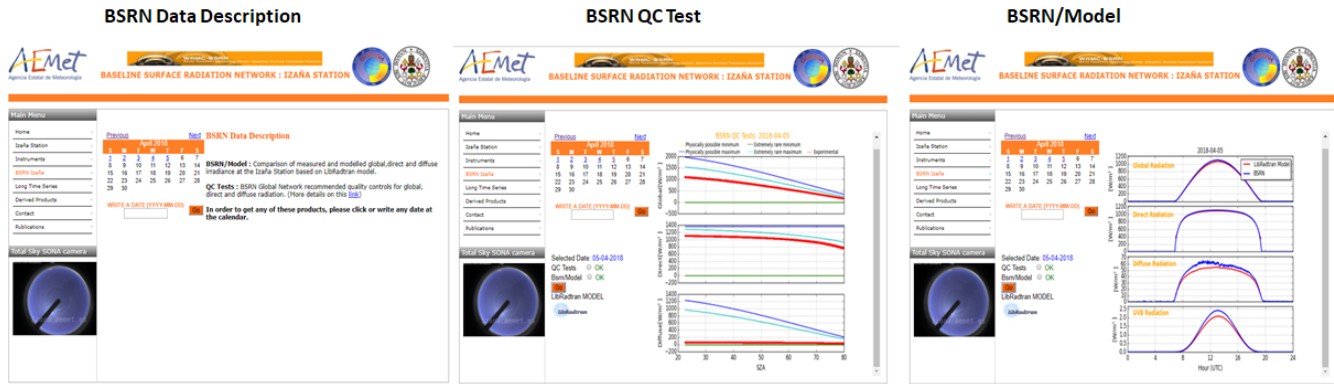

**Figure 12.** BSRN Izaña: BSRN Data Description, QC Test (BSRN Global Network recommended QC for SWD, DIR and DIF) and BSRN/Model comparison for SWD, DIR, DIF and UV-B radiation at IZA using LibRadtran RTM (http://www,bsrn.aemet.es).



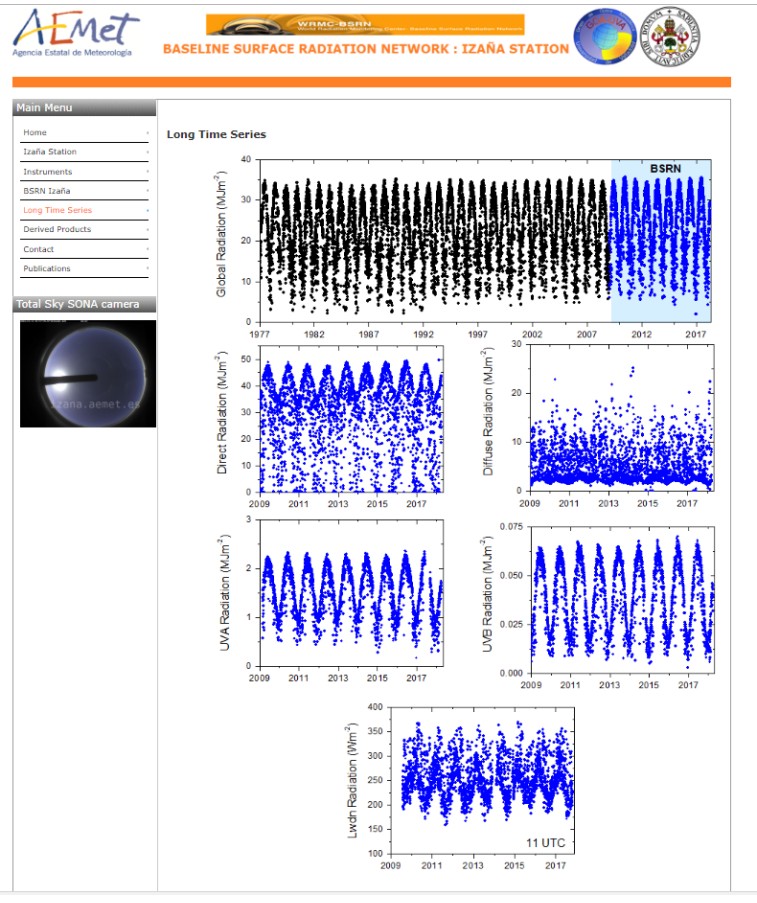

**Figure 13.** IZA time series of daily SWD, DIR, DIF, UV-A and UV-B measurements, and instantaneous LWD data series (11 UTC,) for the period 2009-2017.

pyranometer (PYR) being replaced in 1992 by different instruments (Kipp & Zonen: CM-5, CM-11 and CM-21) (García et al., 2017).

– **Derived products:** From DIR measurements and following the methodology developed by Ellis and Pueschel (1971) the apparent transmission is automatically calculated for the purposes of monitoring clear-sky solar transmission (Figure 14). This apparent transmission is defined as the ratio of the output from a normal-incidence pyrheliometer for a specific pair of SZA corresponding to integer air masses in the morning of a given day:

$$\tau = (I_{dir}/I_{TOA} * sinh)^{1/m_a} \tag{1}$$

where $I_{dir}$ is DIR, $I_{TOA}$ is the top of the atmosphere (TOA) irradiance, $m_a$ is absolute air mass and $h$ is solar elevation angle.



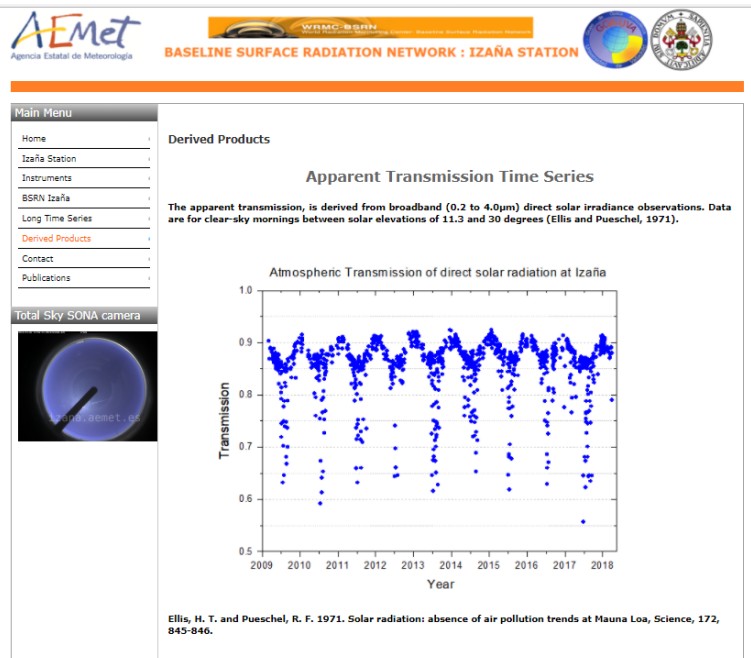

**Figure 14.** Data time series of direct solar radiation atmospheric transmission determined at IZA for the period 2009-2017.

## 4.4 Station-to-BSRN archive file

The last step in the IZA data management is to create the station-to-archive file, that is submitted to the BSRN database on a monthly basis. This procedure is performed using the radiation measurements (Sect. 3.1 and 3.2), radiosonde profiles and total ozone data (Sect. 3.3).



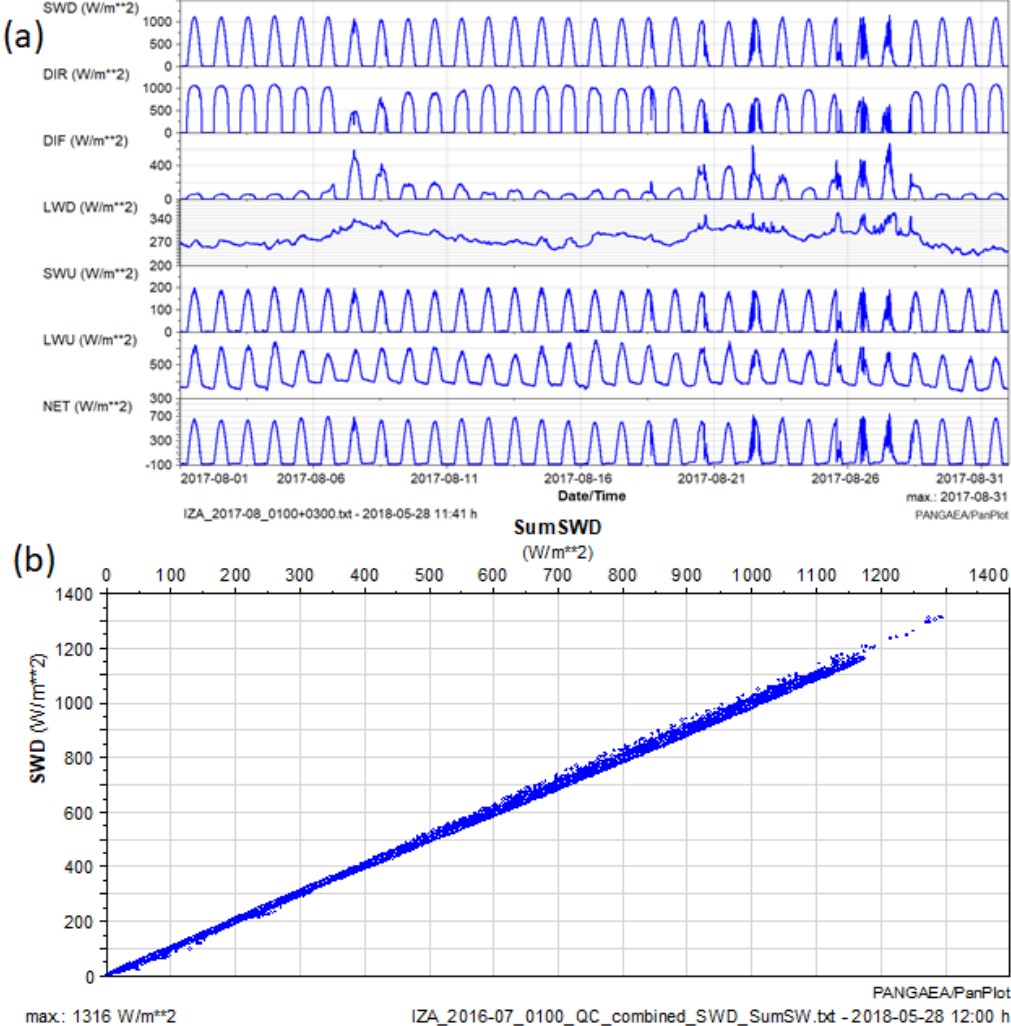

**Figure 15.** Example of (a) visualization of daily data of SWD, DIR, DIF and LWD radiation at IZA on August 2017 (Software: PanPlot; Sieger and Grobe (2013)) and; (b) Intercomparison between SWD on the x-axis and SumSWD (defined as sum of DIF and DIR on a horizontal plane) on the y-axis calculated using the BSRN-Toolbox Schmithüsen et al. (2012).

As shown in Figure 7, the process includes the application of QC tests again to the radiation data using the specific software tools developed by the BSRN. A visual inspection of the monthly data series is made to avoid outliers or detect erroneous data of the different variables before submitting the file to the BSRN database (Figure 15). Finally, if the checks are correct the station-to-archive file is sent by ftp (ftp.bsrn.awi.de).





## 5   Scientific exploitation of IZA BSRN data

A detailed description of the BSRN database has recently been published by Driemel et al. (2018). In this paper some end-users applications of the IZA BSRN data are described.

The IZA BSRN data have been used in diverse research works encompassing several research fields. The most recent publications that have used the IZA BSRN data as part of their work are listed in Table 7, grouped by the research field. Here we only remark on the peer reviewed works but it should be noted that there are many contributions and proceedings that are related also to a greater or lesser extent with the IZA BSRN data.

**Table 7.** Publications that used the IZA BSRN data in the last 5 years.

| Research Field | References |
|---|---|
| Solar spectra determination | Bolsée et al. (2016) |
| Testing of new photocatalytic materials | Borges et al. (2016) |
|  | Borges et al. (2017) |
| Study of ultrafine particle formation processes | García et al. (2014a) |
|  | García et al. (2014b) |
| Reconstruction of historical data series | García et al. (2016) |
|  | García et al. (2017) |
|  | García et al. (2014) |
|  | Linares-Rodriguez et al. (2015) |
| Comparison with simulations from RTMs, | Quesada-Ruiz et al. (2015) |
| Neural Networks | Antonanzas-Torres et al. (2016) |
| and multi-regression techniques | García et al. (2018) |
|  | Urraca et al. (2018) |
|  | Kosmopoulos et al. (2018) |
| Estimating the underlying effective albedo | García et al. (2015) |
|  | Schroedter-Homscheidt et al. (2017) |
| Satellite measurements/forecastings validation | Yu et al. (2018) |
|  | Schroedter-Homscheidt et al. (2018) |
| Climate and long-term studies | Stathopoulos and Matsoukas (2018) |
|  | Lenz et al. (2017) |
| Solar energy application | Hofmann and Seckmeyer (2017) |
|  | Smith et al. (2016) |





In future research, the IZA BSRN data will be essential to analyze accurately the attenuation of different types of clouds in UV, visible and infrared radiation, study the optical and radiative properties of mineral dust, and for solar energy applications, such as solar radiation nowcasting.

## 6 Conclusions

The Izaña station has been part of the BSRN since 2009. IZA BSRN contributes with basic-BSRN radiation measurements: global shortwave radiation (SWD), direct radiation (DIR), diffuse radiation (DIF) and longwave downward radiation (LWD); extended-BSRN measurements: ultraviolet radiation (UV-A and UV-B), shortwave upward radiation (SWU) and longwave upward radiation (LWU); and other measurements: vertical profiles of temperature, humidity and wind obtained from radiosonde (WMO, station #60018) and total ozone column thickness from Brewer spectrophotometer.

Following the recommendations of the BSRN, the quality control tests have been routinely applied. The analysis of the QC results shows a very good data quality that meets the BSRN requirements. The percentage of measurements that are outside the "physically possible" (PPL) and "extreme rare" (ERL) limits is < 1 % for SZA < 90 ° in the period 2009-2017. The ratios between components also provide good results, with > 98 % of measurements within the limits for SZA < 75 °. The poorest result is the SWD/SumSW for 75 ° < SZA < 93 ° with > 96 % of measurements within the defined limits.

In addition we have compared the instantaneous and daily SWD, DIR and DIF measurements with simulations obtained with the LibRadtran RTM. The observed agreement between measurements and simulations is very good: the variance of daily and instantaneous measurements overall agrees within 99% and 98 %, respectively. The simulations underestimate the instantaneous/daily measurements of SWD (-1.68 % / -1.24 %) and DIR (-1.57 % / -1.82 %), while DIF simulations overestimate the instantaneous/daily measurements (0.08 % / 0.84 %). The RMSE is lower than 2.5 % for SWD and DIF for both instantaneous and daily comparisons. These results demonstrated a high consistency between the measurements and simulations reinforcing the reported data quality. The results show also the usefulness of the RTM as a tool for quality control radiation measurements over time.

*Data availability.* The BSRN Izaña radiation measurements are available at http://bsrn.awi.de

*Competing interests.* The authors declare that they have no conflict of interest.

## Appendix A: Solar Radiation Definitions

– **Global Shortwave Radiation (SWD)** : the radiation received from a solid angle of $2\pi$ sr on a horizontal surface in a spectral range from 285 and 3000 nm. The SWD on a horizontal surface is equal to the direct normal radiation times the cosine of the solar zenith angle plus the diffuse irradiance (WMO, 2014).





- **Direct radiation (DIR)** : the radiation measured at the surface of the Earth at a given location with a surface element perpendicular to the Sun in a spectral range from 200 and 4000 nm (WMO, 2014).

- **Diffuse radiation (DIF)** : the radiation measured on a horizontal surface with radiation coming from all points in the sky excluding circumsolar radiation in a spectral range from 285 and 3000 nm (WMO, 2014).

- **Longwave downward radiation (LWD)**: thermal irradiance emitted in all directions by the atmosphere; gases, aerosols, and clouds as received by an horizontal upward facing surface i na spectral range from 4500 to 42000 nm (WMO, 2014).

- **Ultraviolet radiation (UV-B)** : the radiation received from a solid angle of $2\pi$ sr on a horizontal surface spectral range from 280 and 315 nm (WMO, 2014).

- **Ultraviolet radiation (UV-A)** : the radiation received from a solid angle of $2\pi$ sr on a horizontal surface spectral range
from 315 and 400 nm (WMO, 2014).

*Acknowledgements.* This work is part of the activities of the World Meteorological Organization (WMO) Commission for Instruments and Methods of Observations (CIMO) Izaña test bed for aerosols and water vapor remote sensing instruments. Authors thank the BSRN for providing quality control tools and maintaining a centralized quality-assured database. Authors are grateful to Robert P. Stone (NOAA, National Oceanic and Atmospheric Administration) for his audit visit to the Izaña Observatory and corresponding proposal to enroll BSRN, and to
Dr. Ells Dutton (passed away in 2012) for presenting the candidacy of the Izaña station in the 11th Biennial Baseline Surface Radiation Network (BSRN) Scientific Review and Workshop (New Zeland). The careful daily maintenance work made by IZA observers and SIEL-TEC Canarias technicians is very much appreciated. Antonio Cruz, Izaña Atmospheric Research Center computer technician, helped in the development of BSRN. Most of the instruments pictures of this work were provided by Conchy Bayo. Authors appreciate the PMOD/WRC calibration facilities and collaboration. The IZA BSRN program has benefited from results obtained within POLARMOON project funded
by the Ministerio de Economía y Competivdad from Spain, CTM2015-66742-R. We also acknowledge our colleague Dr Celia Milford for improving the English language of the manuscript.





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
