# Peer review of "Description of the Baseline Surface Radiation Network (BSRN) station at the Izaña Observatory (2009-2017): measurements and quality control/assurance procedures"

_Geoscientific Instrumentation, Methods and Data Systems, 2018_

## Referee Comment (RC1) · Anonymous Referee #2 · 15 Nov 2018

The paper gives a very clear overview of the quality control, calibration and measurement methods that are used for the IZA BSRN station.

The paper is very well written and provides information in a high graphical and documented level, the way the station instruments perform. The presented procedures and the web links are very useful for scientists that deal or will deal with these measurements.

I would definitely recommend the publication of this work in GIMDS journal as it includes all the main aspects that the journal is dealing with.

I have to minor comments that could improve the publication.

A comment on the overall instrument's uncertainty (e.g. also in the abstract) would be helpful because future studies of long term changes in the parameters that are measured at this station, would be easy to be justified and assessed by citing this work.

Some additional information on Rayleigh scattering and extraterrestrial LibRadTran model inputs used, could be informative for the reader. I guess for such a low pollution environment the extraterrestrial choice/uncertainty will play an important role on the small differences of model vs measurements reported here.

Other

P1L20 variables

P2L1 contribute

P11L13 A Dep file ?

---

## Author Comment (AC1) · 2 Jan 2019

**Anonymous Referee #2:**

**The paper gives a very clear overview of the quality control, calibration and measurement methods that are used for the IZA BSRN station. The paper is very well written and provides information in a high graphical and documented level, the way the station instruments perform. The presented procedures and the web links are very useful for scientists that deal or will deal with these measurements. I would definitely recommend the publication of this work in GIMDS journal as it in includes all the main aspects that the journal is dealing with.**

*Authors: We appreciate the positive and constructive comments of the Reviewer.*

**I have to minor comments that could improve the publication.**

**A comment on the overall instrument's uncertainty (e.g. also in the abstract) would be helpful because future studies of long term changes in the parameters that are measured at this station, would be easy to be justified and assessed by citing this work.**

*Authors: The instrument uncertainties were included in Sections 3.1 and 3.2 of the original manuscript. However, and to make this information more visible, we have incorporated it into the tables 1 and 2 as follows:*

*Table 1.- Basic-BSRN radiation instruments installed between 2009 and 2017 at IZA BSRN. (SWD, DIR, DIF and LWD). The instruments currently in operation are marked in bold.*

| Parameter | Manufacturer | Type | Serial Number | WRCM | Start Date | End Date | Spectral Range | Instrument uncertainty |
|---|---|---|---|---|---|---|---|---|
| SWD | Kipp & Zonen | CM-21 | 080034 | 61001 | 01/03/2009 | 10/11/2016 | 335-2600 nm | <±1% for daily totals |
| | **EKO** | **MS-802F** | **F15509FR** | **61010** | **11/11/2016** | ---- | **285-3000 nm** | |
| DIR | Kipp & Zonen | CH-1 | 080050 | 61003 | 01/03/2009 | 10/11/2016 | **200-4000 nm** | <±1% for daily totals |
| | **EKO** | **MS-56** | **F15048** | **61012** | **11/11/2016** | ---- | | |
| DIF | Kipp & Zonen | CM-21 | 080032 | 61002 | 01/03/2009 | 10/11/2016 | 335-2600 nm | <±1% for daily totals |
| | **EKO** | **MS-802F** | **F15508FR** | **61011** | **11/11/2016** | ---- | **285-3000 nm** | |
| LWD | Kipp & Zonen | CGR-4 | 080022 | 61004 | 01/03/2009 | 01/05/2009 | 4.5-42µm | < 3% for daily totals |
| | | | 050783 | 61008 | 01/05/2009 | 13/05/2014 | | |
| | | | 080022 | 61004 | 14/05/2014 | 22/07/2014 | | |
| | | | 050783 | 61008 | 23/07/2014 | 30/03/2017 | | |
| | | | 080022 | 61004 | 30/03/2017 | 07/06/2017 | | |
| | | | **050783** | **61008** | **08/06/2017** | ---- | | |

*Table 2. Extended-BSRN radiation instruments installed at IZA BSRN between 2009 and 2017 (UVB, UVA, SWD and LWU). Same as Table 1.*

| Parameter | Manufacturer | Type | Serial Number | WRCM | Start Date | End Date | Spectral Range | Instrument uncertainty |
|---|---|---|---|---|---|---|---|---|
| UV-B | Yankee YES | UVB-1 | 970839 | 61007 | 01/03/2009 | 22/02/2010 | 280-315 nm | <5% daily totals |
| | | | 071221 | 61009 | 22/02/2010 | 22/07/2015 | | |
| | | | 970839 | 61007 | 23/07/2015 | ---- | | |
| UV-A | Kipp & Zonen | UV-A-S-T | 08005 | 61006 | 01/03/2009 | ---- | 315-400 nm | <5% daily totals |
| SWU and LWU | Kipp & Zonen | CRN1 | 030693 | 61005 | 01/03/2009 | 27/11/2016 | PYRA: 305-2800 nm PYRG: 5-50µm | <10% for PYRG and <5% for PYRA daily totals |
| | EKO | MR-60 | S15115.07 | 61013 | 01/01/2017 | ---- | PYRA: 285-3000 nm PYRG: 3-50 µm | |

**Some additional information on Rayleigh scattering and extraterrestrial LibRadTran model inputs used, could be informative for the reader. I guess for such a low pollution environment the extraterrestrial choice/uncertainty will play an important role on the small differences of model vs measurements reported here.**

*Authors:* **As Rayleigh scattering cross secction we have chosen the default provided in the LibRadtran model, that is, the one from Bodhaine et al. (1999). In particular, by default the model uses the equations 22 and 23 of the cited paper.**

**Regarding the extraterrestrial spectrum, we have made a sensitivity test of the LibRadtran model considering three different extraterrestrial spectra (Theakakara, Kuruz and Gueymard). The results obtained are practically the same (Figure 1), so the choice of the extraterrestrial spectrum does not influence the results obtained.**

[Figure]

*Figure 1. Scatterplot of the instantaneous (Wm⁻²) radiation measurements and simulations for (a) global, (b) direct and (c) diffuse radiation considering three extraterrestrial spectra.*

**Following the Referee's comment, the authors have added a new table with input parameters used in the simulations at the manuscript final.**

| Input | Source | Input parameters measured at IZA | References |
|---|---|---|---|
| Radiative transfer equation solver | Disort2 | - | Stamnes et al. (1988,2000) |
| Atmosphere model | Long-term ozonesonde performed at the Botanic Observatory (BTO; Tenerife) | X | Rodriguez-Franco and Cuevas (2013) |
| Solar Flux | Kurucz | - | Kurucz (1992) |
| Ozone cross section | Bass and Paur | - | Bass and Paur (1985) |
| Absorption Parametrization | SBDART | - | (Pierluissi and Peng, 1985); Ricchiazzi et al., 1998 |
| Surface albedo | 0.11 | X | |
| Ozone Column | Brewer spectroradiometer | X | |
| Water vapour column | AERONET products | X | Holben et al. (1998) |
| Aerosol Ångström | AERONET products | X | Holben et al. (1998) |
| Aerosol asymmetry parameter | AERONET products | X | Holben et al. (1998) |
| Aerosol single scattering albedo | AERONET products | X | Holben et al. (1998) |
| Aerosol profile | Shettle | X | Shettle (1989) |
| Solar Zenith Angle | - | - | - |
| Altitude | 2 400 m s.n.m. | - | - |
| Number of streams | 16 | - | - |

*Table 7. The input parameters of the LibRadtran model, their sources, and corresponding references.*

"Following the BSRN recommendations, the detected instantaneous clear-sky periods are simulated and compared with instantaneous and daily radiation measurements. The RTM model used is LibRadtran (http://www.libradtran.org; Mayer and Kylling (2005); Emde et al. (2016)) that has been extensively tested at IZA (García et al., 2014; García et al., 2018). **The measured input parameters used in the LibRadtran model simulations are shown in Table 7** (García et al., 2014; García et al., 2018)…"

**P1L20 variables**

*Authors:* ***Done***

**P2L1 contribute**

*Authors:* ***Done***

**P11L13 A Dep file?**

*Authors:* **The Dep file is that generated after removing mistakes, blank measurements or other errors might cause problems in the data evaluation process. It is referred as "*Dep Data*" in the diagram of Figure 7.**

**References**

Bodhaine, B. A., Wood, N. B., Dutton, E. G., and Slusser, J. R.: On Rayleigh optical Depth calculations, J. Atm. Ocean Technol., 16, 1854–1861, 1999.

---

## Referee Comment (RC2) · Anonymous Referee #4 · 14 Jan 2019

The idea of the paper is to give an overview of different methods used in the IZA BSRN station in the treatment of radiation data and others. The authors explain the quality control, calibration and diverse methodology applied to the data that are submitted to the WRCP. The SWD, DIR and DIF data are compared with LibRadTran radiative transfer model simulations, showing good agreement.

Although comparisons between observed measurements and simulations from clear sky models are a usual work, both description of measurements and instruments at Izaña and methodology applied to data (radiation corrections, quality control) are extended and well defined. The authors present a web site in which the Izaña BSRN station with the associated data is shown, displaying in real time comparisons between measurements and simulations of diverse radiation data (long-term series and derived data). Finally, the authors have included a scientific exploitation of Izaña BSRN data in which they show different works using the data. Therefore, I think the manuscript deserves publication.

————————————————